biomechanics, physiology, biomaterials

epicuticle, lubrication, tribology, leg, articulation, pores

**Author for correspondence:**
Konstantin Nadein
e-mail: k.nadein@gmail.com

# Insects use lubricants to minimize friction and wear in leg joints

Konstantin Nadein[1], Alexander Kovalev[1], Jan Thøgersen[2], Tobias Weidner[2] and Stanislav Gorb[1]

[1]Functional Morphology and Biomechanics, Zoological Institute, Christian-Albrechts University of Kiel, Am Botanischen Garten, 1-9, 24118 Kiel, Germany
[2]Department of Chemistry, Aarhus University, 8000 Aarhus C, Denmark

KN, 0000-0003-3937-5996; AK, 0000-0002-9441-5316; TW, 0000-0002-7083-7004; SG, 0000-0001-9712-7953

A protein-based lubricating substance is discovered in the femoro-tibial joint of the darkling beetle *Zophobas morio* (Insecta). The substance extrudes to the contacting areas within the joint and appears in a form of filiform flows and short cylindrical fragments. The extruded lubricating substance effectively reduces the coefficient of sliding friction to the value of 0.13 in the tribosystem *glass/lubricant/glass*. This value is significantly lower than 0.35 in the control tribosystem *glass/glass* and comparable to the value of 0.14 for the tribosystem *glass/dry PTFE* (polytetrafluoroethylene or Teflon). The study shows for the first time that the friction-reducing mechanism found in *Z. morio* femoro-tibial joints is based on the lubricant spreading over the contacting surfaces rolling or moving at low loads and deforming at higher loads, preventing direct contact of joint counterparts. Besides *Z. morio*, the lubricant has been found in the leg joints of the Argentinian wood roach *Blaptica dubia*.

## 1. Introduction

Lubrication is known as one of the strategies for friction minimization and wear control [1]. In vertebrates, joints are enclosed into a cavity filled out by the synovial fluid serving as a lubricant between contacting cartilage surfaces. These fluid-lubricated joints exhibit a very low coefficient of friction ($\mu$). For example, the coefficient of friction of the stifle-joint in horses in dry conditions was measured as 0.27, and the use of synovial fluid as a lubricant reduced this value to 0.02 [2]. Even lower values of the coefficient of friction were measured for human joints, where they ranged from 0.005 to 0.023 [3], demonstrating the high efficiency of synovial fluid as a lubricant.

The insect body is covered by the integument serving as an exoskeleton composed of fibrous, multi-layered, composite material [4–8]. Typically, it is penetrated by the minuscule pore canals transporting various substances to the epicuticle such as cement and waxes, which are responsible primarily for the protection from desiccation [5–8]. Epicuticular grease, pheromones, fungicidal agents and bactericidal agents are also belong to the cuticular secretions [9–20]. Leg joints of insect exoskeleton are not encapsulated and typical articulation is exposed to the outer space. It has been previously speculated that the epicuticular surface of the contacting areas in a joint is covered with pores that are the openings of canals penetrating the underlying cuticle and being responsible for delivering lubricants towards the contact area [13–21]. However, to date, the friction-reducing mechanism in insect leg joints still remains enigmatic [14,22,23]. We show that insects such as beetles and cockroaches use a semi-solid lubricant to minimize friction in the leg joints.

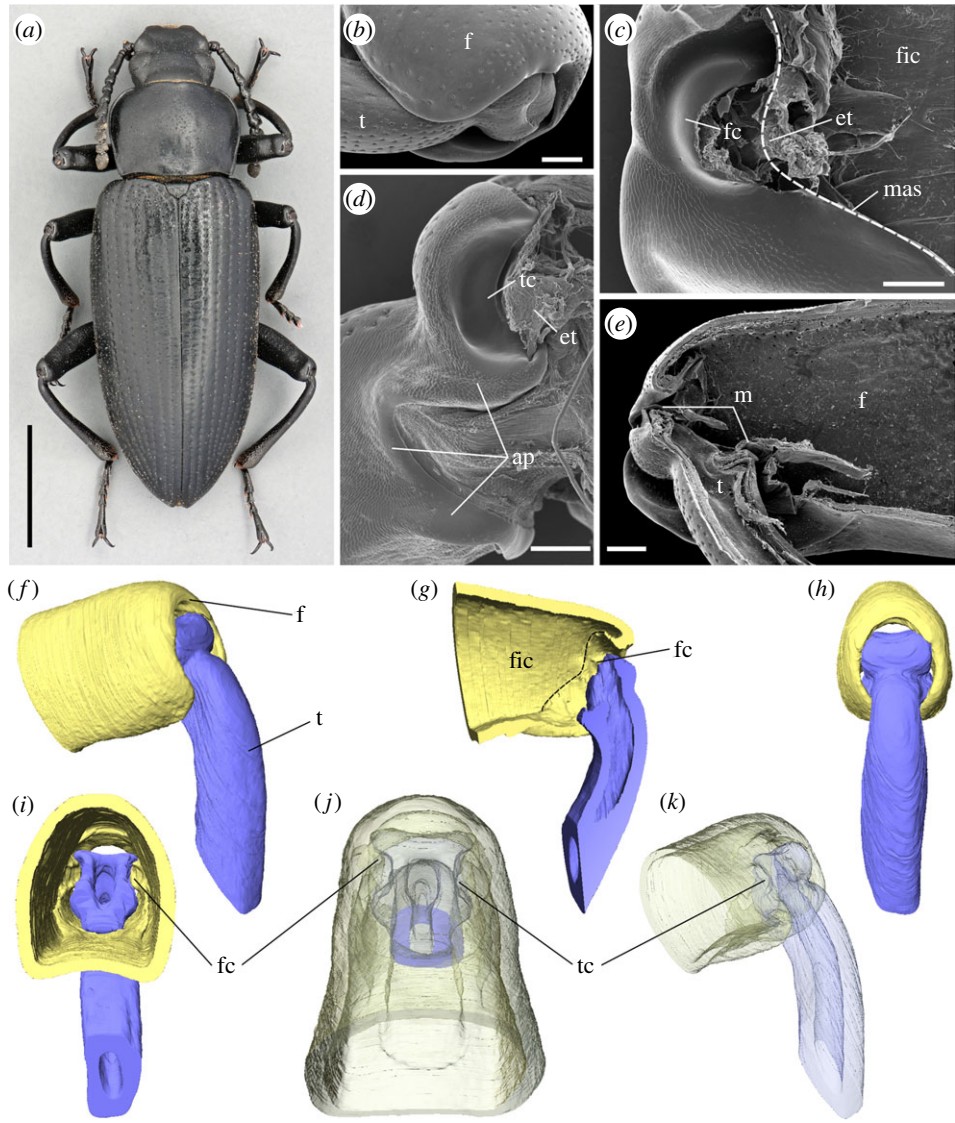

**Figure 1.** Darkling beetle *Zophobas morio* (*a*) and the structure of its femoro-tibial joint (*b–k*). (*b*) Femoro-tibial joint, external appearance. (*c*) Femoral counterpart of the femoro-tibial joint, dashed line indicates the border between femoral internal cavity (fic) and femoral joint counterpart. (*d*) Tibial counterpart of the femoro-tibial joint. (*e*) Femoro-tibial joint, sagittal section. (*b–e*) SEM micrographs. (*f–k*) micro-CT volume reconstructions of the femoro-tibial joint. (*f*) External appearance, antero-lateral view. (*g*) Sagittal section, dashed line indicates the border between femoral internal cavity (fic) and femoral joint counterpart. (*h*), anterior view. (*i*), internal view from the femoral cavity. (*j*) Dorsal view in semi-transparent visualization. (*k*) The same, antero-lateral view. ap, area covered with pores; et, elastic tendon; f, femur; fc, femoral condyle; fic, femoral internal cavity; m, arthrodial membrane; mas, membrane attachment site; t, tibia; tc, tibial concavity. Scale bars, (*a*) 0.5 cm, (*b,c,e*) 200 µm, (*d*) 100 µm. (Online version in colour.)

## 2. Results

### (a) Structure of the femoro-tibial joint

In the darkling beetle *Zophobas morio* (Fabricius, 1776) (Coleoptera: Tenebrionidae) (figure 1*a*), the femoro-tibial joint (figure 1*b*) comprises two counterparts (figure 1*c,d*). The femoral counterpart (figure 1*c,g*) is located at the proximal part of the femur and represented by an open cavity separated from the femoral internal cavity (figure 1*c*) by arthrodial membranes (figure 1*e*). The tibial counterpart is represented by the distal part of the tibia (figure 1*d*) partially inserted into the femoral cavity (figure 1*e,f–k*). The femoral counterpart bears a pair of convex semicircular condyles (figure 1*c,f,g,i*) inserted into corresponding concavities on the tibial counterpart (figure 1*d,f,g,j,k*), allowing rotation along a single axis and connected by elastic tendons (figure 1*c,d*). Both femoral and tibial counterparts are also connected by arthroidal membranes (figure 1*e*). Contacting surfaces of the femoral and tibial counterparts of the joint (figure 1*c,d*) are covered with numerous pore openings.

The pore-bearing area extends nearly to the whole femoral counterpart except the very top of the femoral condyle (figure 1*c*). The area covered by pores contains both smooth and textured regions of the cuticle (figure 2), so that a lot of pores are hidden between the folds of the textured area (figure 2*a,b*). On the tibial counterpart, the pore-bearing area is found on the smooth and textured surfaces proximally from the tibial cavity (figure 1*d*), but not at the bottom of the cavity itself. The average diameter of the pore opening is about 1 µm and a narrow area around each pore is very slightly concave (figure 2*d*). The distance between the pores varies greatly, being usually at least three to four times higher than the diameter of the pore and the total amount of pores is supposed to be hundredfold in a single joint.

### (b) Lubricant appearance

The presence of a substance extruding from the pore openings (figure 2) has been detected by the cryo-SEM for fresh samples of dissected legs and by the conventional SEM for

*Proc. R. Soc. B* **288**: 20211065

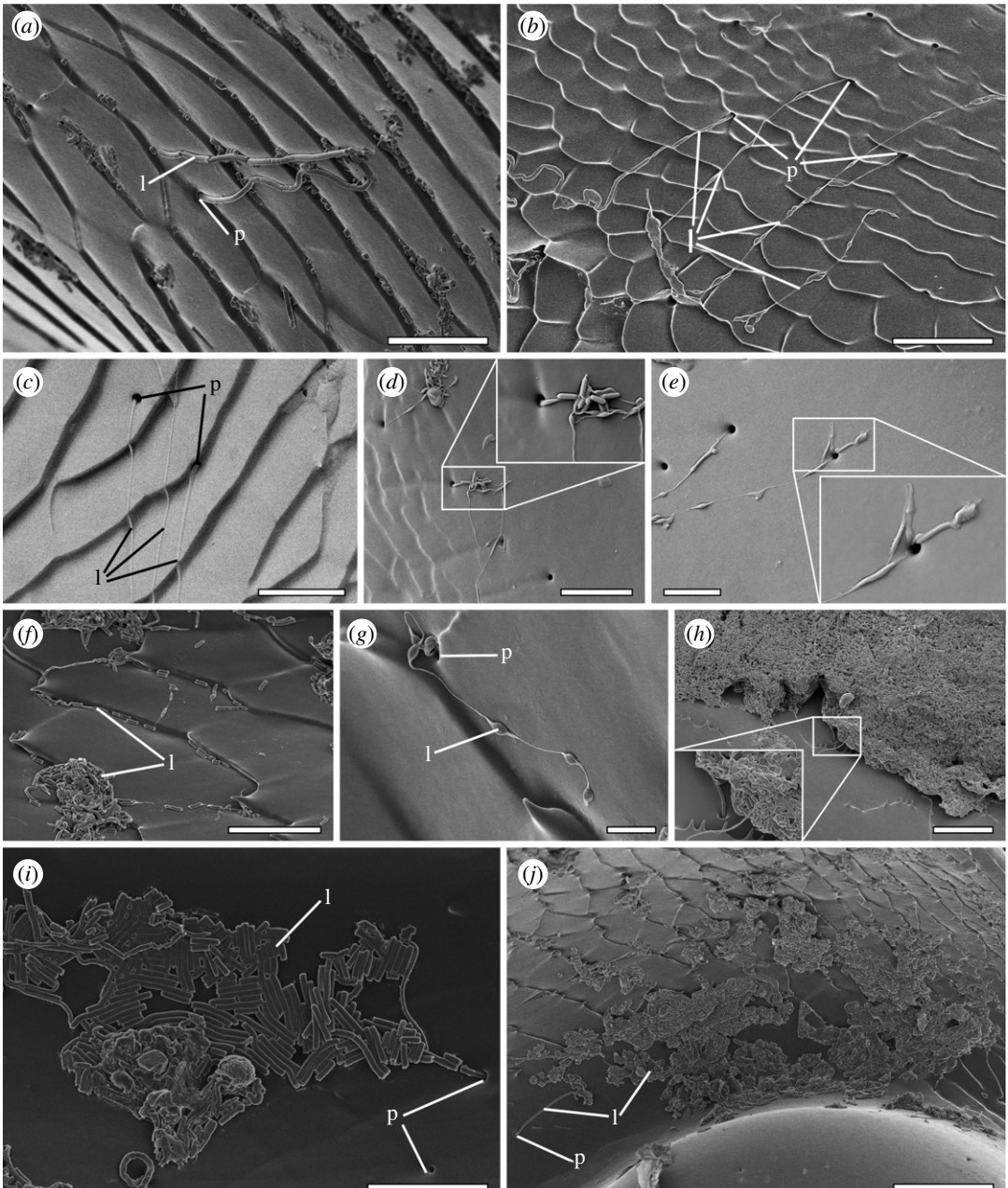

**Figure 2.** Lubricating secretion in the femoro-tibial joint of *Zophobas morio*. (*a*) Lubricant extruding from the pore opening (numerous small fragments over the textured surface of the femoral counterpart are visible). (*b*) Lubricant flows extruding from the pore openings, tibial counterpart. (*c–e*) Diversity of lubricant's appearance, including multiple extruding from the same pore opening, tibial counterpart. (*f,g*) Lubricant flows with a series of thickenings, femoral counterpart. (*h*) Lump of extruded lubricant aggregated on the femoral counterpart at the membrane attachment site. (*i*) Lubricant fragments covering the surface of the tibial concavity. (*j*) Surface at the femoral condyle covering by numerous lubricant fragments. (*a,b,f–j*) SEM micrographs. (*c–e*) Cryo-SEM micrographs. l, lubricant; p, pore opening. Scale bars, (*a–f,i*) 10 μm, (*e*) 5 μm, (*g*) 2 μm, (*h*) 15 μm, (*j*) 25.

dry samples. The substance extruding from the pore opening appears in a form of thick (up to 1 μm, figure 2*a*) or thin (figure 2*b,c*) flows or comparatively short, swollen, cylindrical pieces interconnected by thin bond (figure 2*d–g*). The length of the flows varies significantly from a few micrometres to dozens of micrometres. The flows are often fractured into pieces of different length (figure 2*a,f,i,j*) or remain unbroken reaching a length up to 100 μm. The extruding substance is often produced in an enormous quantity being aggregated into large lumps more than 200 μm long and dozens micrometres thick (figure 2*h*). These lumps usually located on the smooth part of the femoral counterpart close to the membrane attachment site (figure 1*c*).

## (c) Physical and chemical properties of lubricant

At the room temperature (T ≈ 24°C), the substance is insoluble in distilled water, rather weakly soluble in 6 N HCl and chloroform, and soluble in 99.8% ethanol and 10% KOH. A portion of substance left for several days at the room temperature (T ≈ 24°C) on the open air has no signs of evaporation or degradation. The melting point is higher than 100°C.

To determine the molecular composition of the secreted lubricant, we recorded infrared (IR) spectra of non-treated secretion using a micro-sample holder mounted into an ATR-IR instrument. The IR spectrum shown in figure 3 displays all modes expected for proteins. Strong amide A and B resonances near 3250 and 3030 cm$^{-1}$, respectively, are accompanied by amide I and amide II modes near 1650 and 1550 cm$^{-1}$. Strong

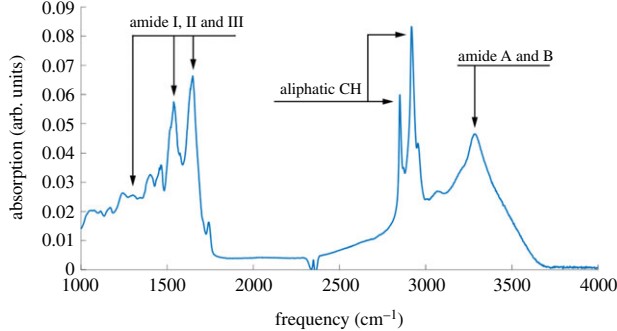

**Figure 3.** The IR spectrum of the secreted lubricant. The spectrum exhibits all resonances expected for a protein-based material. (Online version in colour.)

C–H resonances just below 3000 cm$^{-1}$ clearly show the presence of aliphatic chains as expected for proteins but also fatty acids and a number of other biomolecules. An absorption near 1720 cm$^{-1}$ is indicative of C=O species in the sample, which can be explained by the presence of carboxylic acid-terminated protein side chains but could also indicate the presence of fatty acids or lipids. Considering the relative intensities of the C=O resonance near 1720 cm$^{-1}$ and the amide I mode one can conclude that proteins are, by a large margin, the majority species in the samples. It can therefore be concluded that, unlike many lipid-based substances extruded by insect cuticle, the detected lubricant is substantially protein based. The insolubility in water therefore is not surprising, since many proteins, like keratin, resilin, elastin and collagen, are water-insoluble.

## (d) Tribological experiments

The lubricating property of the extruded substance has been tested in a series of tribological experiments, to examine the influence of the substance on the coefficient of sliding friction (COF, $\mu$). The values of the coefficient of sliding friction have been measured for the tribological system 'glass/extruded substance/glass' and compared to the system 'glass/glass' as a control. For further better comparative understanding, the COF was also measured between two bare glass plates with a similar amount of conventional lubricants: 'glass/dry PTFE/glass' (polytetrafluoroethylene or Teflon) and 'glass/viscous vacuum grease/glass'. These lubricants represent two different types, solid and viscous, respectively, aimed to better understanding the properties of the substance from the insect joints. The glass plates with material in tests between them (or with no material in control) were moving back and forth horizontally at the speed of 78 μm s$^{-1}$ and under the load of 6–44 mN (53.33–766.67 kPa) that corresponded to the traction forces in a single femoro-tibial joint of the beetle moving horizontally (3.75–15.44 mN, corresponding to the pressures of 178.20–731.82 kPa). The experimental results are summarized in the figure 4 and table 1. The COF in the tribosystem glass/extruded substance/glass was on average 0.13. In the control test with glass/glass (without any lubricating substance), the COF was on average 0.35. Rather similar and comparable results were obtained for conventional lubricant PTFE ($\mu = 0.14$). The COF for vacuum grease was significantly higher ($\mu = 1.13$).

Thus, according to results of tribological experiments described above, we assume that the substance extruding in

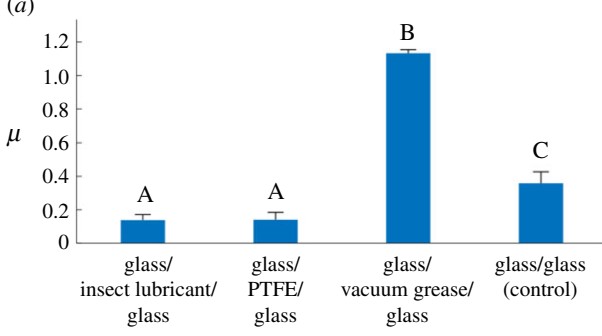

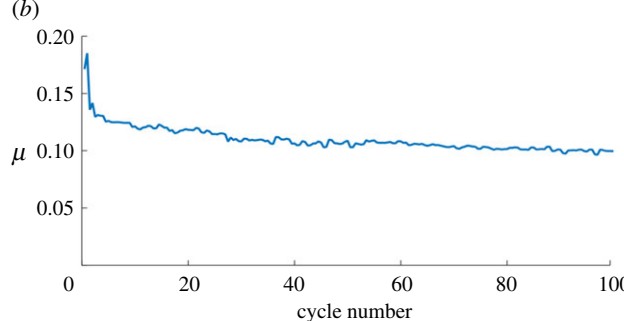

**Figure 4.** Tribological properties of the secretion from the femoro-tibial joint of Zophobas morio. (a) The values of coefficient of friction between cover glass and glass slide without/with various lubricants. Error bars show standard deviations. Values that do not share the same letter are statistically significantly different (one-way ANOVA, $F_{3,13} = 221.6$, $p < 0.001$). (b) Coefficient of friction ($\mu$) dependence on the number of friction cycle (the result of a single experiment as an example). (Online version in colour.)

the femoro-tibial joint of darkling beetle Z. morio effectively reduces friction between contacting surfaces in the joint.

## 3. Discussion

### (a) The friction-reducing mechanism in leg joint of the beetle Zophobas morio

The friction-reducing mechanism in leg joints of the beetle Z. morio is supposed to be the following. The contacting surfaces in the femoro-tibial joint are represented by two primary types of surface profiles: textured (figure 2a,b,e,f,g) and smooth (figure 2c,d). The smooth surface is characteristic of the femoral condyle (figure 1c) and tibial concavity (figure 1d), and also localized along with the membrane attachment site (dividing femoral joint counterpart and femoral internal cavity; figure 1c) of the femoral counterpart. The textured surface covers the rest in both tibial and femoral counterparts (figure 1c,d). There are three types of contacting counterparts in a joint: (i) femur–tibia, i.e. contact between femoral condyle (figure 1c,f) and tibial concavity (figure 1d,f); (ii) arthroidal membrane–tibial (figure 1e); and (iii) arthroidal membrane–femoral counterparts (figure 1e). Obviously, the loads arising in different parts of the joint are not equal. It can be assumed that the maximum loads occur in the contact zone between the femoral condyle and the tibial concavity (figure 1c,d), while the loads in the contact zone between the membrane and the surface of the femoral or tibial counterparts is supposed to be lower. Lubricant spreads over the contacting surfaces by movements of the tibia relative to the femur simultaneously breaking apart into small fragments (figure 2a,i). At low loads and when the gap between contacting surfaces is at least about 1 μm (average diameter

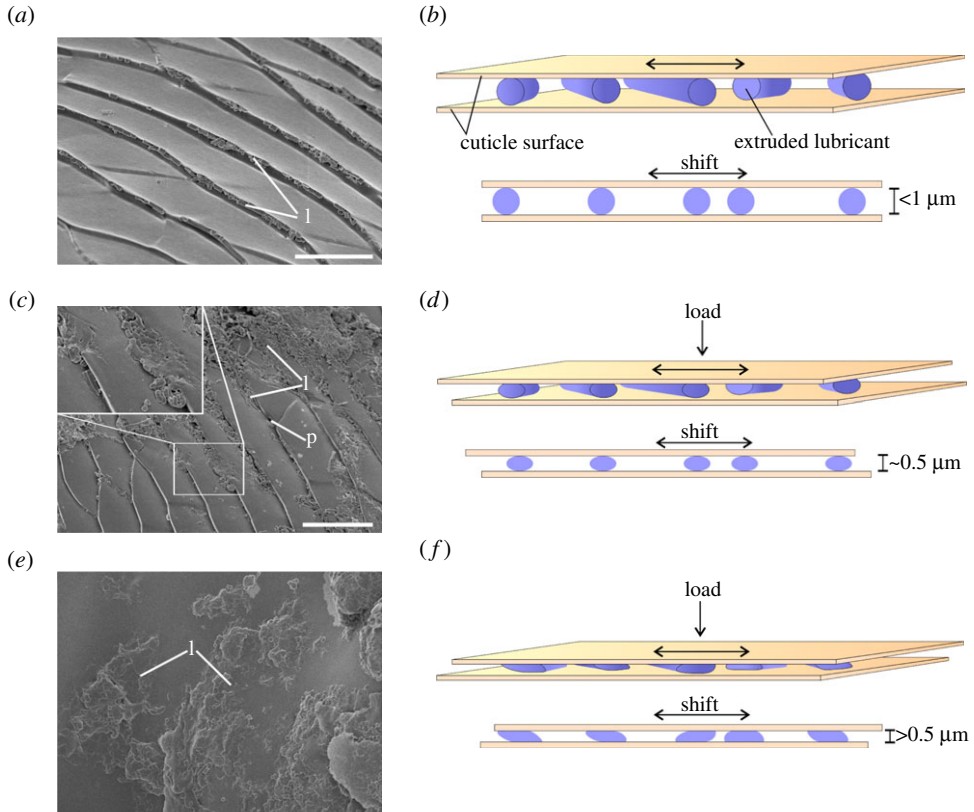

**Figure 5.** SEM visualization of the lubricant (*a,c,e*) and corresponding schematic representations of the lubricant functioning (*b,d,f*). (*a*) The femoral counterpart surface covered by native undeformed fragments of the lubricant, in particular accumulated in the cuticular ridges. (*b*) Fragments of the lubricant, when the distance gap between contacting surfaces is at least 1 μm. (*c*) The femoral counterpart surface covered by partially compressed fragments of the lubricant (see insertion). (*d*) Fragments of lubricant, when the distance gap between contacting surfaces is about 0.5 μm. (*e*) The femoral counterpart surface covered by fragments of lubricant strongly compressed and deformed. (*f*) Fragments of the lubricant, when the distance gap between contacting surfaces is distinctly less than 0.5 μm. l, lubricant; p, pore opening. Scale bars, (*a*) 10 μm, (*c*) 15 μm, (*e*) 20 μm. (Online version in colour.)

**Table 1.** Results of tribological experiments, representing the coefficient of friction (*μ*) for three tested lubricating substances and for the dry control test without any lubricant. Values that do not share the same letter are statistically significantly different.

| material in test | N | μ, range | average | standard deviation | groups |
|---|---|---|---|---|---|
| lubricant from leg joint of the beetle *Zophobas morio* | 8 | 0.10–0.18 | 0.13 | 0.033 | A |
| dry PTFE (Teflon) | 3 | 0.10–0.19 | 0.14 | 0.044 | A |
| viscous vacuum grease (Dow Corning) | 2 | 1.13 | 1.13 | 0.021 | B |
| glass/glass (control) | 4 | 0.26–0.44 | 0.35 | 0.068 | C |

of the thickest pieces of extruded substance), the cylindrical fragments of lubricant may roll over or move on the surface experiencing no or weak deformations (figure 5*a,b*). At high loads, when the distance gap between joint counterparts is smaller than 1 μm, the pieces of lubricant start experiencing plastic deformations, get thinner and spread over the surface (figure 5*c–f*). Thus, the lubricant prevents direct physical contact between contacting surfaces and may absorb shock energy during sudden loads. Being easily subjected to plastic deformations, it is presumably adaptive to the surface texture and loads. At low loads, the substance behaves like a solid, i.e. functions as a dry lubricant. At higher loads and corresponding plastic deformations, the lubricant behaves as a semi-solid or viscous substance.

Due to its presence between contacting surfaces, the lubricant prevents direct physical contact between them and thus presumably reduces wear. The peculiar property of the lubricating substance is its susceptibility to fragmentation (figures 2*a,f,i* and 5*a,c*). Being broken apart in numerous small fragments, a relatively small amount of the extruded substance may homogeneously cover large cuticle areas (figure 2*j*). This property allows lubricant to create a thin layer covering the large area between the surfaces in a joint, and, moreover, small fragments may more easily penetrate into small gaps and prevent in this way building of concentrated load. Such a homogeneous lubricant distribution may effectively reduce adhesion and static friction in the joint.

## (b) Lubricant in the leg joints of other insects

Lubrication-based friction-reducing mechanism in leg joints is found to be common in beetles. The substance, extruding from the pore openings in the femoro-tibial joint, has been found in Congo rose chafer *Pachnoda marginata* (Drury,

1773) (Scarabaeidae) electronic supplementary material, figure S1), ground beetle *Carabus coriaceus* Linnaeus, 1758 (Carabidae) (electronic supplementary material, figure S2), lesser silver water beetle *Hydrochara caraboides* (Linnaeus, 1758) (Hydrophilidae) (electronic supplementary material, figure S3*a,b*) and the coxo-trochanteral joint of carrion beetle *Oiceoptoma thoracicum* (Linnaeus, 1758) (Silphidae) (electronic supplementary material, figure S3*c,d*). The pores in the femoro-tibial joint have also been found in the tiger-beetle *Cicindela campestris* Linnaeus, 1758 (Cicindelidae) (electronic supplementary material, figure S3*e,f*) and click-beetle *Agrypnus murinus* (Linnaeus, 1758) (Elateridae) (electronic supplementary material, figure S4), suggesting possible presence of the lubricant within their joints. In general, we assume the significant universality of this friction-reducing mechanism among insects evidenced by the discovery of the lubricant also in the femoro-tibial joints of Argentinian wood roach *Blaptica dubia* (Serville 1838) (Blattodea: Blaberidae) (electronic supplementary material, figure S5) that is phylogenetically quite distant from beetles.

## (c) Comparison of lubricant-based friction-reducing mechanisms in the joints of vertebrates and insects

Reducing joints' friction is a challenge in both vertebrates and arthropods. Being structurally very different the joints of these animal groups evolved a common lubricant-based solution for friction/wear reduction. Their friction-reducing mechanisms have some similarities and differences. In vertebrates, the friction minimization by means of synovial fluid is based on the boundary lubrication that works synergistically with other mechanisms, such as pressurization-driven elastohydrodynamic lubrication [1,24]. The mechanism of friction reduction discovered in the femoro-tibial joint of darkling beetle *Z. morio* is rather different and apparently based on the semi-solid or highly viscous lubricant dispersed over the contacting surfaces by the plenty of minute fragments. However, separating the contacting surfaces by means of a lubricant and preventing their physical solid-solid contact (which is typical for most lubricants) is the main similarity in the mechanisms of vertebrates and insects. The use of a solid, semi-solid or highly viscous substance as a lubricant in insects is presumably associated with the opened type of the joints, since the presence of liquid lubricant in an open joint would inevitably lead to its constant losses and the need for production in significant quantities.

## (d) Perspectives

It is well known that friction and wear are critical issues limiting the operational lifetime and negatively influencing technological potential. That is why joints and hinges in various engineering mechanisms need lubrication. However, conventional lubrication in micro-joints in microelectromechanical systems (MEMS) cannot always be applied due to the strong stiction between counterparts. Moreover, environmental conditions and/or modes of operation are often variable, which places high demands on the multi-adaptability of lubricants, while most of them, though highly effective, are usually strongly specialized. The structure of locomotor organs of insects, in particular their joints, strongly resembles mechanical micro-joints and artificial micro-hinges. Having similar challenges, insects evolved an interesting solution to reduce friction in their leg joints. In addition, insolubility of the herewith discovered lubricant in water allows insects to use it in a highly humid environment, which supports the idea of its multi-adaptability.

Insects are traditional objects in biomimetic research including their locomotory systems for robotics application [25]. Further studies on the properties of the discovered lubricant may be of interest as a promising source of ideas for further biomimetic applications in the area of novel lubricating materials. In this regard, this research may be of particular interest for robotics and MEMS technology, and especially for prosthetics, in order to develop a new generation of completely bio-organic lubricants with friction-reducing properties similar to PTFE (Teflon).

Data accessibility. The data are provided in the electronic supplementary material [26].

Authors' contributions. K.N.: conceptualization, funding acquisition, investigation, methodology, writing—original draft, writing—review and editing; A.K.: formal analysis, investigation, methodology, writing—review and editing; J.T.: formal analysis, investigation, writing—review and editing; T.W.: formal analysis, investigation, methodology, writing—review and editing; S.G.: conceptualization, methodology, supervision, writing—review and editing. All authors gave final approval for publication and agreed to be held accountable for the work performed therein.

Competing interests. We declare we have no competing interests

Funding. This work was supported by the grant for K.N. from the Deutsche Forschungsgemeinschaft (temporary position for principal investigators, NA 126472-1). T.W. received financial support from Danmarks Frie Forskningsfond (DFF FNU 8021-00046B).

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
