## [Peer Review File · Proceedings of the Royal Society B: Biological Sciences]

Review History

RSPB-2021-1065.R0 (Original submission)

Review form: Reviewer 1

Recommendation

Accept with minor revision (please list in comments)

Scientific importance: Is the manuscript an original and important contribution to its field?

Good

General interest: Is the paper of sufficient general interest?

Good

Quality of the paper: Is the overall quality of the paper suitable?

Good

Is the length of the paper justified?

Yes

Should the paper be seen by a specialist statistical reviewer?

No

Do you have any concerns about statistical analyses in this paper? If so, please specify them explicitly in your report.

No

It is a condition of publication that authors make their supporting data, code and materials available - either as supplementary material or hosted in an external repository. Please rate, if applicable, the supporting data on the following criteria.

Is it accessible?

Yes

Is it clear?

Yes

Is it adequate?

Yes

Do you have any ethical concerns with this paper?

No

Comments to the Author

Please see the attached file. (See Appendix A)

Review form: Reviewer 2

Recommendation

Major revision is needed (please make suggestions in comments)

Scientific importance: Is the manuscript an original and important contribution to its field?

Acceptable

General interest: Is the paper of sufficient general interest?

Acceptable

Quality of the paper: Is the overall quality of the paper suitable?

Acceptable

Is the length of the paper justified?

Yes

Should the paper be seen by a specialist statistical reviewer?

No

Do you have any concerns about statistical analyses in this paper? If so, please specify them explicitly in your report.

No

It is a condition of publication that authors make their supporting data, code and materials available - either as supplementary material or hosted in an external repository. Please rate, if applicable, the supporting data on the following criteria.

Is it accessible?

N/A

Is it clear?

N/A

Is it adequate?

N/A

Do you have any ethical concerns with this paper?

No

Comments to the Author

Briefly speaking, the article in its short main text is novel and interesting, but the reviewer asks them to understand that the femoral cuticle in beetles is an other construction then in other insects, but they can not repeat in the manuscript any novel details described by the reviewer.

Review form: Reviewer 3 (Bo Persson)

Recommendation

Accept with minor revision (please list in comments)

Scientific importance: Is the manuscript an original and important contribution to its field?

Good

General interest: Is the paper of sufficient general interest?

Good

Quality of the paper: Is the overall quality of the paper suitable?

Good

Is the length of the paper justified?

Yes

Should the paper be seen by a specialist statistical reviewer?

No

Do you have any concerns about statistical analyses in this paper? If so, please specify them explicitly in your report.

No

It is a condition of publication that authors make their supporting data, code and materials available - either as supplementary material or hosted in an external repository. Please rate, if applicable, the supporting data on the following criteria.

Is it accessible?

Yes

Is it clear?

Yes

Is it adequate?

Yes

Do you have any ethical concerns with this paper?

No

Comments to the Author

This is a very interesting study and I recommend publication.

However, I want the authors to add some more information about the friction experiment:

Add to the main paper the sliding speed and the nominal contact pressure (the information is now only in the supplementary materials).

Also add information about what these quantities are for the insect.

Some insects (and the tree frog) use fluids for adhesion and in those cases the fluid is likely pulled spontaneously into the contact area by capillary effects. But if the lubricant in the present case is in a solid state how is it injected into the leg joints in the narrow pore channels?

It appears to me that this cannot occur spontaneously by capillary effects but may require a big force or pressure which is generated how?

Decision letter (RSPB-2021-1065.R0)

04-Jun-2021

Dear Dr Nadein

I am pleased to inform you that your manuscript RSPB-2021-1065 entitled "Insects use lubricants to minimize friction and wear in leg joints" has been accepted for publication in Proceedings B.

The referee(s) have recommended publication, but also suggest some minor revisions to your manuscript. Therefore, I invite you to respond to the referee(s)' comments and revise your manuscript. Because the schedule for publication is very tight, it is a condition of publication that you submit the revised version of your manuscript within 7 days. If you do not think you will be able to meet this date please let us know.

- 1) A text file of the manuscript (doc, txt, rtf or tex), including the references, tables (including captions) and figure captions. Please remove any tracked changes from the text before submission. PDF files are not an accepted format for the "Main Document".
- 2) A separate electronic file of each figure (tiff, EPS or print-quality PDF preferred). The format should be produced directly from original creation package, or original software format. PowerPoint files are not accepted.
- 3) Electronic supplementary material: this should be contained in a separate file and where possible, all ESM should be combined into a single file. All supplementary materials accompanying an accepted article will be treated as in their final form. They will be published

alongside the paper on the journal website and posted on the online figshare repository. Files on figshare will be made available approximately one week before the accompanying article so that the supplementary material can be attributed a unique DOI.

It is a condition of publication that data supporting your paper are made available either in the electronic supplementary material or through an appropriate repository. Please see our Data Sharing Policies <https://royalsociety.org/journals/authors/author-guidelines/#data>.

If you wish to submit your data to Dryad (<http://datadryad.org/>) and have not already done so you can submit your data via this link [http://datadryad.org/submit?journalID=RSPB&manu=\(Document not available\)](http://datadryad.org/submit?journalID=RSPB&manu=(Document not available)) which will take you to your unique entry in the Dryad repository. If you have already submitted your data to dryad you can make any necessary revisions to your dataset by following the above link. Please see <https://royalsociety.org/journals/ethics-policies/data-sharing-mining/> for more details.

Sincerely,
Dr Locke Rowe
mailto:proceedingsb@royalsociety.org

Associate Editor
Comments to Author:

The discovery that some insects produce lubricants analogous to human synovial fluid is fascinating. Reviewer 1 asks for clarification of the generality of the results to insects and a longer more contextual discussion on epicuticular grease and insect secretions. Reviewer 2 also asks

about the generality of this work, considering the cuticle of beetles may be different from other insects. Reviewer 3 asks for more details on the lubrication experiment and discussion of how semi-solid materials can be ejected through the holes. Note that Reviewer 1 and 2 have word and pdf files with additional comments. Overall, some toning down of the scope of the results may be needed and some greater context should be given with regards to other insect secretions. All reviewer comments should be addressed.

Reviewer(s)' Comments to Author:

Referee: 1

Comments to the Author(s)

Please see the attached file.

Referee: 2

Comments to the Author(s)

Briefly speaking, the article in its short main text is novel and interesting, but the reviewer asks them to understand that the femoral cuticle in beetles is an other construction then in other insects, but they can not repeat in the manuscript any novel details described by the reviewer.

Referee: 3

Comments to the Author(s)

This is a very interesting study and I recommend publication.

However, I want the authors to add some more information about the friction experiment:

Add to the main paper the sliding speed and the nominal contact pressure (the information is now only in the supplementary materials).

Also add information about what these quantities are for the insect.

Some insects (and the tree frog) use fluids for adhesion and in those cases the fluid is likely pulled spontaneously into the contact area by capillary effects. But if the lubricant in the present case is in a solid state how is it injected into the leg joints in the narrow pore channels?

It appears to me that this cannot occur spontaneously by capillary effects but may require a big force or pressure which is generated how?

Author's Response to Decision Letter for (RSPB-2021-1065.R0)

See Appendix B.

Decision letter (RSPB-2021-1065.R1)

09-Jun-2021

Dear Dr Nadein

I am pleased to inform you that your manuscript entitled "Insects use lubricants to minimize friction and wear in leg joints" has been accepted for publication in Proceedings B.

Your article has been estimated as being 7 pages long. Our Production Office will be able to confirm the exact length at proof stage.

Data Accessibility section

Open Access

Paper charges

Sincerely,

Appendix A

Review on RSPB-2021-1065

Insects use lubricants to minimize friction and wear in leg joints

by Konstantin Nadein, Alexander Kovalev, Jan Thøgersen, Tobias Weidner & Stanislav Gorb

A long-term mystery uncovered – the lubrication of beetle leg joints! Congratulations to the authors on a meaningful, well-illustrated manuscript!

Several minor issues to be raised ...

The Title “Insects use lubricants to minimize friction and wear in leg joints” does not fit the provided study and obtained results. This was done, and which is shown: “Darkling beetles use lubricants to minimize friction and wear in leg joints”. – In this context, the 6 other beetle and a cockroach species mentioned in the Supplementary Material are rather confusing and not referred to in the main body of the manuscript.

The Introduction is succinct, however, quite short-cut. In particular, related to arthropod joints and cuticle one to three further statements could be included, such as about epicuticular grease and insect secretion in general. This aspect in comparison with the present results would still emphasize the significance of the author’s findings. Also, the special structures covering the joint and/or joint edge surfaces could be considered, discussing the “multifunctionality” of the joints unifying lubrication on the one hand and friction enhancement on the other hand – this role of the structures on the joint cuticle counterparts is also worth to be considered in illustrations and discussion.

The question arises about the state, condition, and age of the tested animals. Do aged ones secrete a similar lubricant and similar amounts?

The structure of secretion, in particular in Fig. 1d, f, I looks unusual compared to other viscoelastic fluids found in insects and plants so far. They remember bacteria-shaped particles; maybe characteristic for proteinaceous material as detected for the joint lubrication? – Here some comparison with the state of literature about shapes of proteinaceous structures would be value-adding and supporting the innovation of the present study.

Detailed comments:

page 1, Abstract

A protein-based

of the darkling beetle

The extruded lubricating

That the friction-reducing

found in *Z. morio* femoro-tibial joints

Redundant: surfaces in the form of numerous minor fragments of cylindrical shape rolling

page 1, Keywords: epicuticle, lubrication, tribology, leg, head, articulation, pores

page 2, Introduction

that the (epi)cuticular surface

page 2, Results

(Fabricius, 1776) (Coleoptera, Tenebrionidae)

The pore-bearing area

The area covered by pores

that a plenty of pores is hidden

The average diameter of the pore opening is about 1 μm and surrounded by a narrow, very slightly concave area.

The presence of a substance

a length up to

page 4, Results

Please provide the distinct values for “At the room temperature” and “at the room temperature”.

spectra of non-treated secretion in-situ on leg joints using

is (substantially/remarkably) protein-based – Mostly doesn't fit the rather qualitative results.

collagen, to name a few, are

The lubricating property of

better comparative understanding

page 5, Results

that the substance extruding

page 5, Discussion

(a) The friction-reducing mechanism

The friction-reducing mechanism in leg joints of the beetle *Z. morio*

the distance gap between joint counterparts

Please specify "it": "It allows" – What allows?

page 5, Discussion, (b)

Here, several examples are given for the presence of a lubricant in insect leg joints. – This fact should be mentioned in the Introduction to avoid the implication of a new finding which has been already previously supported.

page 11, Author contributions

tribological experiments

analyzed the lubricant

Table 1

average = mean?

Figure 3. The infrared spectrum of the secreted lubricant. The spectrum exhibits all resonances expected for a protein-based material. – And, what else? Here, all the compounds should be mentioned and indicated to underline the chemical complexity of the secretion mixture.

Figure 4 would benefit from schematic insets illustrating the experimental setups because the MM part is not ad-hoc accessible in the present paper style.

Figure 4
The values of coefficient of friction

Figure 4 ... are statistically significantly different. – Please provide the related statistical data: one-way ANOVA? $F = \dots$, $P = \dots$

Figure 5
of the lubricant, in particular accumulated in the cuticular ridges.

Figure 5 could benefit from some more distinct labels, e.g., relating the images in a, c, and e to the schemes in b, d, and f (arrows, colors, or similar ...)

page 18

Which individual, which species – all species mentioned in (a)?: Legs from a freshly CO₂-anesthetised individual

How, for how long the beetles were kept under which distinct conditions?

no sublimation?: ... into the liquid nitrogen and sputter-coated

dry foreleg of which species?

Lubricant samples of which species?

How in detail the samples of lubricant have been collected? – instruments, procedure, how long before the analyses, etc.? How much volume?

p. 20

direction-independent

78 $\mu\text{m s}^{-1}$

p. 21

Which forces in the femoro-tibial joint? Friction forces?

Obtained force values

p. 22

$F_{3,13} = 221.6, P < 0.001$

Figure S1-5 would benefit from insets indicating the body position where the images were obtained and which counterparts meet how each other.

p. 28

Supplementary Material

Fig. S5

Appendix B

Response to Referees

Associate Editor
Comments to Author:

The discovery that some insects produce lubricants analogous to human synovial fluid is fascinating. Reviewer 1 asks for clarification of the generality of the results to insects and a longer more contextual discussion on epicuticular grease and insect secretions.

Reviewer 2 also asks about the generality of this work, considering the cuticle of beetles may be different from other insects.

Reviewer 3 asks for more details on the lubrication experiment and discussion of how semi-solid materials can be ejected through the holes.

Note that Reviewer 1 and 2 have word and pdf files with additional comments. Overall, some toning down of the scope of the results may be needed and some greater context should be given with regards to other insect secretions. All reviewer comments should be addressed.

Reviewer(s)' Comments to Author:

Referee: 1
Comments to the Author(s)
Please see the attached file.
Authors: The corrections have been made.

Referee: 2
Comments to the Author(s)
Briefly speaking, the article in its short main text is novel and interesting, but the reviewer asks them to understand that the femoral cuticle in beetles is an other construction then in other insects, but they can not repeat in the manuscript any novel details described by the reviewer.
Authors: The femoral cuticle in beetles can be of different construction. However the study of the femoral cuticle in beetles and its comparison with that of other insects was out of the scope of the present paper. In this study we aimed to describe the newly discovered lubrication-based friction-reducing mechanism and check its effectiveness experimentally. We believe that the possible difference in the structure of the femoral cuticle cannot affect the generality of the work since the lubricant is found in the cockroach species as well.

Referee: 3
Comments to the Author(s)
This is a very interesting study and I recommend publication.
However, I want the authors to add some more information about the friction experiment:
Add to the main paper the sliding speed and the nominal contact pressure
(the information is now only in the supplementary materials).
Authors: The information is added.

Also add information about what these quantities are for the insect.
Authors: The information is added.

Some insects (and the tree frog) use fluids for adhesion and in those cases the fluid is likely pulled spontaneously into the contact area by capillary effects. But if the lubricant in the present case is in a solid state how is it injected into the leg joints in the narrow pore channels? It appears to me that this cannot occur spontaneously by capillary effects but may require a big force or pressure which is generated how?
Authors: This is a very interesting question. At the moment the mechanism of delivery the viscous or semi-solid lubricant to the cuticular surface in unclear. In this first paper about lubrication-based friction-reducing mechanism we tried to avoid any speculations on this but focused on the experimental checking of the lubricative properties of the extruded substance. Undoubtedly this question will be addressed in our further studies of the lubricant in the insects' joints.

Review on RSPB-2021-1065

Insects use lubricants to minimize friction and wear in leg joints

by Konstantin Nadein, Alexander Kovalev, Jan Thøgersen, Tobias Weidner & Stanislav Gorb

A long-term mystery uncovered – the lubrication of beetle leg joints! Congratulations to the authors on a meaningful, well-illustrated manuscript!

Several minor issues to be raised ...

The Title “Insects use lubricants to minimize friction and wear in leg joints” does not fit the provided study and obtained results. This was done, and which is shown: “Darkling beetles use lubricants to minimize friction and wear in leg joints”. – In this context, the 6 other beetle and a cockroach species mentioned in the Supplementary Material are rather confusing and not referred to in the main body of the manuscript.

Authors: Thank you very much! Indeed, the experimental part of the manuscript deals with the darkling beetle *Zophobas*. However, the lubricant was found in cockroaches and other beetles as well, not in the darkling beetle only. The presence of the lubricant in the leg joints of other insect species studied is a fact that we found significant to publish. Beetles and cockroaches are insects (actually from two major groups – Hemimetabola and Holometabola) and therefore to the large extent can represent Insecta. In our opinion, restriction the title to the only darkling beetle may inevitably (and wrongly) suggest that only this insect species possesses lubricant. There are also other reasons in favour of the chosen title. First, the reason, why only the lubricant from legs of darkling beetle *Zophobas* was tested, is that it was the only one possible to collected in a sufficient amount. The lubricant of other insects is produced in smaller amounts that were impossible to collect and test. Perhaps in the future, with involvement of other equipment, it will be possible to collect and test the lubricants from the other insects. Second, the electron microscopy data on 6 other beetles and a cockroach support the hypothesis of the broad (or even universal) distribution of the lubricating-based friction-reducing mechanism in the insects’ leg joints. It means it is not an occasional, unique or deviant phenomenon found in the darkling beetle *Zophobas* only. Third, the title initiates and promotes the further interest in studying lubricant-based mechanism of friction-minimisation in joints of insects that for a long time has been overlooked.

The Introduction is succinct, however, quite short-cut. In particular, related to arthropod joints and cuticle one to three further statements could be included, such as about epicuticular grease and insect secretion in general. This aspect in comparison with the present results would still emphasize the significance of the author’s findings.

Authors: We provided short general statements about cuticle, epicuticular grease and insect secretion in general.

Also, the special structures covering the joint and/or joint edge surfaces could be considered, discussing the “multifunctionality” of the joints unifying lubrication on the one hand and friction enhancement on the other hand – this role of the structures on the joint cuticle counterparts is also worth to be considered in illustrations and discussion.

Authors: We did not mention ‘multifunctionality’ but ‘multi-adaptability’ which is discussed in the manuscript. Indeed, the contacting surfaces in joints are of different structure from the smooth to variously textured. In fact, the variety of textured surfaces is huge from species to species (or at least from genus to genus). Therefore, it seems not possible to characterise all this in detail in such a short paper. The ‘speciality’ and role of these structures in leg joints is out of the scope of the present paper. Nevertheless, the tribological properties of the joints’ surfaces are currently in the focus of our current project.

The question arises about the state, condition, and age of the tested animals. Do aged ones secrete a similar lubricant and similar amounts?

Authors: We did not test the lubricant production or properties regarding to the age, state and/or condition of animals since it was out of the scope of our study. The tested adults were mature enough (at least two or three weeks or elder). Lubricant production varies greatly from individual to individual and even from leg to leg of the same individual. It may supposedly be connected with the physiological state of an individual, its physical activity, and many other reasons which are awaiting for study.

The structure of secretion, in particular in Fig. 1d, f, l looks unusual compared to other viscoelastic fluids found in insects and plants so far. They remember bacteria-shaped particles; maybe characteristic for proteinaceous material as detected for the joint lubrication? – Here some comparison with the state of literature about shapes of proteinaceous structures would be value-adding and supporting the innovation of the present study.

Authors: Here in this paper we aimed to report on the lubricating-based mechanism in insect joints for the first time. The structures under consideration are definitely not bacteria: one of the co-authors has some

experience in SEM studies on different kinds of bacteria. We are not sure, which literature about which proteinaceous structures is suggested.

Detailed comments:

Authors: All the corrections highlighted blue by the Reviewer are made.
Here further are the comments requiring the responses.

page 4, Results

Please provide the distinct values for “At the room temperature” and “at the room temperature”.

Authors: The value of the room temperature is added.

is (substantially/remarkably) protein-based – Mostly doesn't fit the rather qualitative results.

Authors: The ATR-FTIR analysis is indeed qualitative and we believe is an appropriate way to understand the principal chemical composition of the lubricant at the present state of knowledge and study.

Quantitative analysis at the moment is hard to accomplish since the amount of lubricant is quite small for the most of the available methods. Nevertheless, based on the intensity of the resonances it is possible to suggest on the relative abundance of this or that types of molecules. According to the obtained spectra one can conclude that the set of peaks is a clear ‘fingerprint’ of proteins and the other compounds (if any) are in the significantly smaller amount undetectable by this very sensitive method.

Please specify “it”: “It allows” – What allows?

Authors: Changed to ‘This property’.

page 5, Discussion, (b)

Here, several examples are given for the presence of a lubricant in insect leg joints. – This fact should be mentioned in the Introduction to avoid the implication of a new finding which has been already previously supported.

Authors: Yes, there are reports on the presence of some amorphous substances in the insect joints, but these have never been structurally and experimentally studied. There were only suppositions (as ideas or hypotheses) without factual observational evidences or experiments. The insertion ‘such as beetles and cockroaches’ is added to the Introduction.

Table 1

average = mean?

Authors: yes, it is arithmetic mean.

Figure 3. The infrared spectrum of the secreted lubricant. The spectrum exhibits all resonances expected for a protein-based material. – And, what else? Here, all the compounds should be mentioned and indicated to underline the chemical complexity of the secretion mixture.

Authors: All the components that were possible to detect by the ATR-FTIR are mentioned in the text. They are presented by the peaks of different intensity and assigned to resonances that were identified as different components of proteins and, presumably, lipids. Such a set of peaks at the specific resonances bands is a ‘fingerprint’ of the specific type of substance as proteins in this case.

Figure 4 would benefit from schematic insets illustrating the experimental setups because the MM part is not ad-hoc accessible in the present paper style.

Authors: Unfortunately the very limited space of the PRSB paper style prevents us from increasing of the current number or size of figures as well as inclusion of the Materials and Methods.

Figure 4

The values of coefficient of friction

Authors: Corrected.

Figure 4 ... are statistically significantly different. – Please provide the related statistical data: one-way ANOVA? $F = \dots$, $P = \dots$

Authors: The statistical data are added.

Figure 5 could benefit from some more distinct labels, e.g., relating the images in a, c, and e to the schemes in b, d, and f (arrows, colors, or similar ...)

Authors: Corrections were made.

Which individual, which species – all species mentioned in (a)?: Legs from a freshly CO₂-anesthetised individual

Authors: the legs from a freshly CO₂-anesthetised individuals of *Zophobas morio*, *Pachnoda marginata* and *Blaptica dubia*. Correction is made.

How, for how long the beetles were kept under which distinct conditions?

Authors: The beetles *Zophobas morio* and *Pachnoda marginata* and a cockroach *Blaptica dubia* were kept in a standard culture conditions for these insects. There are numerous descriptions in the Internet available. The time of longevity of every individual was not registered and varied from weeks to months.

no sublimation?: ... into the liquid nitrogen and sputter-coated

Authors: No, it is not.

dry foreleg of which species?

Authors: The fore leg of *Zophobas morio* was taken. The corrections is made.

Lubricant samples of which species?

Authors: The lubricant of *Zophobas morio* was taken. The corrections is made.

How in detail the samples of lubricant have been collected? – instruments, procedure, how long before the analyses, etc.? How much volume?

Authors: The lubricant lumps was taken manually from the dissected femoral joint counterparts by the clean and thin needle and put into the minute clean plastic vials. The approximate total volume of lubricant was about 0.04 mm³. The analysis has been performed within a week after collecting. Since the lubricant does not undergo degradation or evaporation we suppose that its chemical composition did not change. The value of the volume of lubricant is added in the text.

Which forces in the femoro-tibial joint? Friction forces?

Authors: Traction forces. The corrections is made.

Figure S1-5 would benefit from insets indicating the body position where the images were obtained and which counterparts meet how each other.

Authors: The figure plates S1-5 depict the surfaces of the leg joints taken from the dissected legs. The samples are placed on the SEM stub to be visible under the different angles. It is hard to specify the body position for every single image. The principal structure of the leg joint of a beetle exemplified by *Zophobas morio* is shown in the Figure 1. The structure of the leg joints of other beetles is similar in such details as femoral condyle, tibial concavity, etc., but the specific shape varies from taxon to taxon.

p. 28

Supplementary Material

Fig. S5

Authors: Done.